# Effects of Regular Brazil Nut (*Bertholletia excelsa* H.B.K.) Consumption on Health: A Systematic Review of Clinical Trials

**DOI:** 10.3390/foods11182925

**Published:** 2022-09-19

**Authors:** Alessandra da Silva, Brenda Kelly Souza Silveira, Brenda Vieira Machado de Freitas, Helen Hermana M. Hermsdorff, Josefina Bressan

**Affiliations:** 1Laboratory of Energy Metabolism and Body Composition, Department of Nutrition and Health, Universidade Federal de Viçosa, Viçosa 36570-900, MG, Brazil; 2Institute of Public Policies and Sustainable Development, Universidade Federal de Viçosa, Viçosa 36570-900, MG, Brazil

**Keywords:** functional foods, metabolic risk factors, nuts, nutrition, oilseeds, selenium

## Abstract

The Brazil nut (BN) is a promising food due to its numerous health benefits, but it is still necessary to systematically review the scientific evidence on these benefits. Thus, we examined the effects of regular BN consumption on health markers in humans according to the health state (with specific diseases or not) of the subjects. PubMed, Embase^®^, and Scielo databases were used to search for clinical trials. The PRISMA guideline was used to report the review, and the risk of bias for all studies was assessed. Twenty-four studies were included in the present review, of which fifteen were non-randomized. BNs were consumed in the context of a habitual free-living diet in all studies. Improvement in antioxidant status through increased levels of selenium and/or glutathione peroxidase activity in plasma, serum, whole blood, and/or erythrocytes was observed in all studies that evaluated antioxidant status, regardless of the health state of the sample. In addition, healthy subjects improved lipid markers and fasting glucose. Subjects with obesity had improvement in markers of lipid metabolism. Subjects with type 2 diabetes mellitus or dyslipidemia improved oxidative stress or DNA damage. Subjects undergoing hemodialysis benefited greatly from BN consumption, as they improved lipid profile markers, oxidative stress, inflammation, and thyroid function. Older adults with mild cognitive impairment improved verbal fluency and constructional praxis, and controversial results regarding the change in a marker of lipid peroxidation were observed in subjects with coronary artery disease. In conclusion, the benefits of BN consumption were found in different pathways of action and study populations.

## 1. Introduction

The Brazil nut (BN) (*Bertholletia excelsa* H.B.K.) is an edible seed of the Brazil nut tree native to the Amazonia biome. According to data provided by International Nut Dried Fruit (INC), Bolivia is the main producer of BNs, followed by Peru and Brazil. The world consumption of BNs, considering a total supply of BNs minus ending stock, was 26,250 tons in 2020/2021 [1]. Like other nuts such as walnuts, hazelnuts, macadamia, pistachios, and almonds, several bioactive compounds are also present in BNs [2]. Unsaturated fats, minerals, vitamins, fibers, and phytochemicals give nuts potential and recognized beneficial health effects [3]. The BN contains smaller amounts of protein compared to plant-based protein sources such as tropical vegetables; however, in 100 g of BNs, 14.3 g is from protein. Although small doses of BNs are recommended daily due to the selenium (Se) content (depending on the growing area, one unit is sufficient to reach the recommended daily intake of Se), BNs can be a potential alternative source of protein in diets [4]. However, unlike other nuts, the BN is the richest source of selenium (Se), a key element in forming antioxidant defense systems, modulation of the immune system, and helping to prevent ageing-related diseases [5].

In 2017, Cardoso et al. revised the health benefits of BNs and addressed their composition, safety, planting, processing, and use [3]. One study regarding BN safety and toxicity showed the absence of health risks regarding the presence of aflatoxins and radioactivity in 30 samples of BN originating from the Amazon [6]. In addition to the heavy metal barium, other elements may be present in BNs, such as strontium. However, the forms and levels of these elements in BNs do not seem harmful to human health [3,7,8]. Other review articles on BNs are also available. However, they have specific objectives and do not provide the most current findings of all studies that explore the effects of regular BN consumption on human health [3,7,8,9,10,11]. Hence, we aim to carry out a systematic review regarding the effects of regular BN consumption on human health. We also discussed the mechanisms involved in these effects and critically assessed available literature and future perspectives on nutrition science.

## 2. Materials and Methods

### 2.1. Protocol and Registration

This systematic review was carried out per the “Preferred Reporting Items for Systematic Reviews and Meta-Analyses (PRISMA)” guidelines [12] and was registered in PROSPERO (https://www.crd.york.ac.uk/prospero/, accessed on 11 September 2022), registration number CRD42021198965.

### 2.2. Literature Search

We identified studies by searching the following electronic databases: MEDLINE/PubMed (https://pubmed.ncbi.nlm.nih.gov/, accessed on 11 September 2022), Embase^®^, and Scielo. The descriptors used were based on Medical Subject Headings (MeSH) (Appendix A Table A1). We conducted an exhaustive literature review using “intervention” from PICOS (population/intervention/comparator/outcome/study design) search criteria to identify all studies with the BN in any health outcome.

Filters were used to select studies on humans and clinical trials when available. We also searched for articles on this theme in the reference list of previously published review articles [3,7,8,9,10,11]. Table A1 lists the search terms used. The last search was conducted in March 2022. The search was performed independently by two authors (AS and BKSS). First, titles and abstracts were selected. The articles were then read in full, and eligible studies were selected. Any disagreement between authors was resolved by consensus. Duplicate articles were removed using Mendeley^®^ software.

### 2.3. Eligibility Criteria

The following eligibility criteria were considered to answer our research question, “What are the effects of regular BN consumption on human health?”

(i).Original clinical trials, randomized or not, controlled or not.(ii).Studies that evaluated the effects of regular BN consumption on any human health marker.(iii).Studies with any dose and time of intervention.(iv).Studies in which BNs were consumed as whole seed.(v).Studies evaluating any health marker, anthropometrics, or body composition and those examining lipid, glucose, kidney, liver, and other markers.(vi).Studies with any subject profile (healthy subjects or subjects with comorbidities), and with any sex.

The following exclusion criteria were applied:(i).Postprandial studies.(ii).Studies with children or animals, observational designs, reviews, congress abstracts, letters, protocol articles, notes, and in vitro analyses.(iii).Studies that did not investigate the effects of the BN on human health markers(iv).Interventions that supplemented BNs with minerals/vitamins or other nutritional enhancements.(v).Interventions with BN oil or flour.(vi).Interventions that included behavioral modifications, such as physical activity.

Studies with children were not considered due to different Se requirements compared to other ages.

### 2.4. Selecting Studies and Data Extraction

Two independent authors (AS and BKSS) selected the studies by analyzing the titles, abstracts, and full texts. Disagreements between the authors were resolved by consensus. In the absence of a full article or when additional information was needed to compile the results, an email was sent to the corresponding author requesting the article or information. Two independent authors (AS and BKSS) extracted the following data from the eligible studies:(i).Name of the first author, year of publication, and country.(ii).Sample characteristics (number of participants, presence of diseases, age, and body mass index (BMI).(iii).Characteristics of the intervention (description of each intervention group and the doses of BNs used with their Se contents).(iv).Study design and duration.(v).Markers evaluated in the study.(vi).Observed results.

We grouped the markers evaluated by the studies into major categories: antioxidant status (referred to Se, glutathione peroxidase (GPx), and/or SELENOP levels analyzed in plasma, serum, erythrocytes), lipid metabolism markers (high-density lipoprotein (HDL)-c, low-density lipoprotein (LDL)-c, total cholesterol (TC), triglycerides (TGs), TC/HDL-c, LDL-c/HDL-c, very-low-density lipoprotein (VLDL)-c, paraoxonase 1, Apo-A1, and Apo B), inflammatory markers (C-reactive protein (CRP), tumor necrosis factor (TNF), interleukins (IL), tool-like receptors, nuclear factor kappa B (NF-κB)), anthropometry and body composition markers (weight, BMI, waist circumference, and fat mass), glucose metabolism markers (glucose, hemoglobin A1c (HbA1c), and Homeostatic Model Assessment for Insulin Resistance (HOMA-IR)), thyroid function markers (thyroid-stimulating hormone (TSH) and triiodothyronine (T3)), cognitive function markers (a consortium to establish a registry for Alzheimer’s disease, verbal fluency, Boston naming test, and constructional praxis), and oxidative stress or DNA damage markers (oxidized (ox)LDL-c, 8-Epi-Prostaglandin F2 alpha (8-epi-PGF2α), malondialdehyde, nuclear factor erythroid 2-related factor 2 (Nrf2), nicotinamide adenine dinucleotide phosphate (NAD(P)H): quinone oxidoreductase 1 (NQO1), thiobarbituric acid reactive substances (TBARS), and 8-hydroxy-2′-deoxyguanosine(8-OHdG)).

### 2.5. Bias Risk Assessment

Two authors (AS and BVMF) independently assessed the risk of bias, following the Joanna Briggs Institute (JBI) Reviewer’s Manual. This appraisal assesses methodological quality and determines the extent to which a study has addressed the possibility of bias in its design, conduct, and analysis. Thirteen questions for randomized clinical trials and nine questions for non-randomized clinical trials were answered for each study included in the systematic review. The answers to these questions were classified as yes, no, unclear, or not applicable [13]. A third author (BKSS) resolved disagreements.

## 3. Results and Discussion

### 3.1. Study Selection

We identified 463 records and removed 27 duplicates. During the titles and abstracts screening, 404 articles were excluded because they did not meet the eligibility criteria. Thirty-two articles remained for full-text evaluation. However, eight were excluded. Postprandial studies (*n* = 2), abstract congress (*n* = 1), unavailable articles (*n* = 1), intervention with granulated BN flour (*n* = 3), and child studies (*n* = 1) were excluded. Eleven articles were revised [3]. As result, 24 articles were incorporated per the eligibility criteria (Figure 1). Table 1 and Table 2 present the characteristics of these studies.

### 3.2. Studies’ Characteristics

Approximately 93% of the publications were from research conducted in Brazil between 2008 and 2022. Thirteen [17,19,20,21,22,26,31,32,33,34,35,36,37] of the twenty-four studies included in this systematic review are new compared to the review carried out by Cardoso et al. (2017).

Of the twenty-four, nine publications were the results of randomized clinical trials. Of these, two studies were placebo-controlled [14,15], seven were controlled [16,17,18,19,20,21,22], and sixteen were non-randomized clinical trials. Only two non-randomized studies were controlled [23], whereas all others were uncontrolled [14,15,24,25,26,27,28,29,30,31,32,33,34,35,36,37]. Regarding the evaluation of biases using the checklist proposed by the JB, 64% of the answers were “yes”, 15% were “no”, 16% were “unclear”, and 5% were “not applicable”. This means that most of the studies evaluated in this review had a low risk of bias and, therefore, met the methodological requirements of study design, conduct, and analysis (Figure 2).

Regarding sample characteristics, studies involving healthy subjects or those with comorbidities, such as obesity, chronic kidney disease, mild cognitive impairment (MCI), type two diabetes, dyslipidemia, and coronary artery disease, were included (Figure 3). We highlight that the health classification indicated in this systematic review is similar to that of other researchers. For example, Thomson et al. (2008) and Strunz et al. (2008) included overweight subjects in their analyses. However, we included the study in the category of healthy subjects based on the eligibility criteria described by the authors of the original articles. In other words, we evaluated subjects without diseases diagnosed by their respective studies.

BNs (whole seed) were consumed in a free-living habitual diet in all studies. The Se concentration in BNs varied from 48 µg in six units [18] to 1.261 µg only in one unit [19].

Legend: GPx, glutathione peroxidase; HDL-c, high-density lipoprotein cholesterol; SELENOP, selenoprotein P; RBCV, red blood cell velocity; TC, total cholesterol; LDL-c, low-density lipoprotein cholesterol; TG, triglycerides; oxLDL-c, oxidized low-density lipoprotein cholesterol; DNA, deoxyribonucleic acid; MDA, malondialdehyde; SOD, superoxide dismutase; BMI, body mass index; WC, waist circumference; Nrf2, nuclear factor erythroid 2-related factor 2; NQO1, NAD(P)H: quinone oxidoreductase 1; T3, triiodothyronine; FT4, free thyroxine; TNF, tumor necrosis factor; interleukin 6; 8-OHdG, 8-hydroxydeoxyguanosine; NF-κB, nuclear factor kappa B; TBARS, thiobarbituric acid reactive substances.

### 3.3. BNs Consumed by Healthy Subjects (Subjects without Diagnosed Diseases)

Regular (15 days to 12 weeks) consumption of BNs (1–11 units/day; 48 µg to 862.65 µg of Se) improved the antioxidant statuses (increased Se in plasma and erythrocytes, GPx, GPx3, SELENOP, and SELENOP mRNA expression), and lipid profiles (decreased TC and increased HDL-c cholesteryl reception) of healthy subjects and reduced their fasting glucose [16,17,21,30,32,38] (Figure 3). Similarly, acute consumption of 1–10 units/day of BNs (156 µg to 1560 µg of Se) associated with a normocaloric diet improved the antioxidant statuses (increased Se in plasma) and lipid profiles (increased HDL-c, decreased LDL-c, and atherogenic indices) in 24 postprandial hours up to 30 days after a single day of consumption [38,39]. Considering that the recommended plasma Se range is between 60–100 µL, only one study with healthy subjects observed that the participants were Se-deficient at baseline (Figure 4A). In this study, Se-deficient subjects displayed an increased reception to cholesteryl esters by HDL-c after BN consumption [24].

Oxidative stress and related diseases, such as cardiovascular diseases (CVDs), are favored when antioxidant status is inferior to oxidizing agents in the body. Considering the importance of primary health care for disease prevention and health promotion, including BNs in the diet can help maintain the antioxidant status equilibrium by Se and related proteins [40,41]. In response to BN consumption, increased plasma SELENOP [34] and its expression [18] were observed in healthy subjects, confirming that SELENOP levels are a biomarker of Se status. Furthermore, an increase in GPx activity in the plasma and whole blood was observed in adult New Zealanders after 12 weeks, indicating the importance of Se in the production of antioxidant enzymes [14].

There is no evidence concerning the mechanisms involved in the beneficial effects of BN consumption on human lipid profiles. However, in experimental studies, Se supplementation downregulated Apo B and 3-hydroxy-3-methylglutaryl coenzyme A (HMG-CoA) reductase expression in hypercholesterolemic rats [42]. This finding may potentiate a beneficial effect of BN consumption on the lipid profile, as HMG-CoA participates in the cholesterol biosynthesis pathway, while Apo B is the main protein component of LDL-c. In another experimental study, co-supplementing rats fed a high-fat diet with Se and magnesium decreased blood and liver TG levels and TC/HDL-c and TG/HDL-c ratios while increasing antioxidant enzymes. Furthermore, co-supplementation inhibited hepatic lipogenesis gene expression, decreased HMG-CoA reductase, and increased cholesterol 7α-hydroxylase (CYP7A1) and lecithin cholesterol acyltransferase (LCAT) in the liver [43]. LCAT removes cholesterol from the blood and tissues and may help explain the increase in HDL-c and cholesteryl ester reception by HDL-c after BN consumption.

BN consumption decreased fasting glucose levels in healthy subjects [32]. In a study of non-diabetic subjects, supplementation with 200 µg of Se (as Se yeast) for six weeks reduced glycated hemoglobin but not fasting glucose [44]. In an observational study of participants with adequate Se status, a 10 μg/L increase in Se was associated with a 1.5% increase in insulin and a 1.7% increase in HOMA-IR [45]. Se status is an important determinant of the effect that Se supplementation will have on the body. In contrast, pistachio consumption showed beneficial effects on glucose metabolism markers in prediabetic subjects. The authors hypothesized that a synergistic effect of unsaturated fats, polyphenols, and carotenoids in the food matrix improved insulin sensitivity through the PI3K-AKT pathway [46].

Some studies have also explored the effect of polymorphisms in specific genes on the responses of healthy subjects to BN consumption, highlighting the influence of genetics on food consumption responses. TC decreased in the GA + AA and GG groups in subjects with polymorphism rs7579 in the SELENOP1 gene. However, allele A carriers tended to have higher cholesterol levels after consumption, mainly after four weeks of intervention. In addition, allele A carriers of polymorphism rs3877899 in the SELENOP1 gene had lower glucose levels at baseline and after eight weeks of intervention [32]. T allele (CT + TT) of the rs1050450 polymorphism in the GPx1 gene tended to have lower GPx activity than CC carriers. In contrast, subjects with polymorphism rs7579 in the SELENOP1 gene had increased plasma SELENOP1 concentrations after eight weeks of intervention, but with no difference between the GG and GG + AA alleles [34]. GPx1 mRNA expression increased in subjects with the CC genotype for rs1050450 after BN consumption, while it did not change in T allele carriers. SELENOP mRNA expression was higher in allele A carriers for rs7579 than in GG subjects after BN consumption [33,34]. SELENOS and SELENOF mRNA expression levels remained unchanged after BN consumption.

### 3.4. BN Consumption by Subjects with Obesity

Different effects were found in subjects with obesity who consumed BN depending on the amount of Se present in the BN. Consumption of three–five units/day of BNs (mean Se: 108.5 µg) increased serum Se levels compared to baseline. TC, TGs, and ox-LDL levels reduced after BN consumption, while red blood cell velocity increased compared to a placebo in adolescents with obesity after 16 weeks of consumption (Figure 3). No changes were observed in anthropometric and glucose metabolism markers, CRP, GPx-3, and 8-epi-PGF2α levels [15]. Another study with women with obesity observed an improvement in plasma Se, GPx activity, and HDL-c, and a decrease in atherogenic indices after participants consumed one unit/day of BN (290 µg of Se) for eight weeks [27]. However, the BNs had higher Se contents compared to the study performed by Maranhão et al. (2011) [15], and the subjects were Se-deficient at baseline (Figure 4B). Moreover, improvements in plasma Se and GPx activity were observed in the Pro/Pro, Pro/Leu, and Leu/Leu genotypes of the Pro198Leu glutathione peroxidase polymorphism. In contrast, DNA damage decreased only in Pro/Pro genotype carriers versus the baseline, which was significantly lower than that in carriers of the Leu/Leu genotype [26].

As previously discussed, no human studies have explained the relationship between Se and lipid metabolism. However, animal studies have noted a reduction in HMG-CoA reductase after Se supplementation and consequently decreased endogenous cholesterol production [42]. The BN is a food matrix and, in addition to Se, also contains unsaturated fats, such as monounsaturated fatty acids (MUFAs), which have recognized functions in TC, TGs, and LDL-c in healthy subjects [47,48]. The benefits of the BN in reducing ox-LDLs, known oxidative stress marker and a risk factor for atherosclerosis [49], are easier to explain because Se is a non-enzymatic antioxidant and contributes to the selenoproteins’ formation, adding to the redox balance by scavenging free radicals [50]. Excessive concentrations of these free radicals in the body can oxidize LDL-c, stimulating inflammation and leading to endothelial dysfunction in the intimal artery.

Weight and BMI did not decrease after regular BN consumption in subjects with obesity [15], suggesting that BN consumption cannot modulate weight when consumed in the context of a habitual diet, without energy restriction, for example. In addition, weight and BMI also did not increase. Studies have shown that despite the high-calorie density and high-fat content of nuts, habitually consuming nuts is not related to weight gain [51].

On the other hand, despite an increase in antioxidant status markers, women with obesity supplemented with one unit/day of BN containing approximately three times the upper limit (UL) of Se (400 µg) for eight weeks had increased pro-inflammatory cytokine expression and decreased GPx1 expression compared to the control group. Furthermore, increased miR-454-3p and miR-584-5p expression levels were observed [19,20]. The findings of this study reinforce that Se deficiencies and overload are harmful to health. It has been hypothesized that excessive Se consumption may play a pro-oxidant role by inducing reactive oxygen species (ROS), activating the Akt-NF-κB pathway, and further stimulating leukocytes and pro-inflammatory cytokine genes [52]. Simultaneously, the decrease in GPx expression can be explained by the fact that from a specific concentration of Se in the blood, GPx activity reaches its maximum [53]. Moreover, these studies note the beneficial health effects of consuming BN with Se contents below the tolerable UL for this population with high adiposity, reinforcing the need for attention regarding the choice of BN consumed based on Se content.

In summary, improvements in antioxidant status and lipid metabolism markers were observed after the minimal consumption of BNs containing an average Se of 108.5 µg in subjects with obesity (Figure 3). Se doses above the UL have shown adverse effects in women with obesity.

### 3.5. BN Consumption by Subjects with Dyslipidemia, Type 2 Diabetes, or Coronary Artery Disease

An increase in inflammatory cytokines and oxidative stress is expected in dyslipidemia, diabetes mellitus, and CVD etiology and progression. In this context, Se deficiency is related to an increased risk of CVDs and mortality, possibly due to reduced selenoprotein synthesis and antioxidant capacity feeding back inflammation [54,55].

BN is a source of Se, MUFAs, polyphenols, and phytochemicals. However, a single unit of BN does not contain significant amounts of MUFAs and bioactive compounds. The nutrient that stands out in its composition is the Se because a single unit of BN can contribute five times more than the daily requirement for this mineral [10]. For this reason, we believe that most of the benefits achieved from BN intake are related to Se content (Table A2).

Consumption of one unit/day of BN (290 µg of Se) for 24 weeks improved antioxidant statuses (increased GPx activity and Se in the plasma and erythrocytes and reduced MDA and creatine kinase) in subjects using statins, while SELENOP mRNA expression was not affected [36] (Figure 3). Statins inhibit cholesterol synthesis via the mevalonate pathway, an important pathway for selenoprotein synthesis. Therefore, subjects who use statins are at an increased risk of developing Se deficiency and oxidative stress [56,57]. BN is a source of Se and may help maintain the nutritional status and oxidative balance of Se in patients using statins. Despite being instructed to maintain their usual diet, these subjects had reduced body weights and BMI. It is unlikely that the reductions in body weight and BMI were related to BN consumption. The authors reported that the effect of Se on BMI occurred under suboptimal Se conditions, which was not the case for the participants [36]. In addition, caloric restriction is necessary for weight loss, regardless of Se intake. However, studies have shown a relationship between selenium and adipogenesis [58], including increased lean mass and muscle mass [59]. Regarding the presence of polymorphisms, BN consumption increased GPX1 mRNA expression only in subjects with the rs1050450 CC genotype. SELENOP mRNA expression was significantly lower in subjects with the rs7579 GG genotype before and after the intervention.

No significant changes in Nrf2, NF-κB, NQO1, or peroxisome proliferator-activated receptor (PPARβ/δ) mRNA expression levels or markers of lipid metabolism and inflammation were found in subjects with coronary artery disease after three months of intervention with one unit/day of BNs (~290.5 µg of Se) compared to the control group. Controversial results regarding changes in TBARS were observed in the two articles on the effect of BN consumption in individuals with coronary artery disease [21,22] (Figure 3). The authors did not provide the baseline values of plasma Se values. This missing information is important to clarify the discussion because subjects with a normal Se status may not benefit from Se supplementation [55,60]. Another hypothesis is that these patients had multiple metabolic disorders and, therefore, required higher Se doses or extended interventions to benefit from BN consumption.

The consumption of one unit/day of BN (213.67 µg of Se) increased serum Se and decreased DNA damage ex vivo in subjects with type 2 diabetes treated with oral glucose-lowering medication and insulin [35] (Figure 3). Again, BN consumption had benefits related to oxidative stress in a different population and was closely associated with Se status and selenoprotein synthesis.

Nevertheless, the subjects had increased waist circumferences and fasting blood glucose levels compared to the baseline, but the HbA1c levels remained unaltered [35]. Plasma Se and fasting glucose levels did not correlate. Considering that all the subjects were overweight and instructed to maintain their regular diet, these results can be attributed to the unbalanced diet. BN consumption was unable to improve glucose metabolism markers. Other nuts, especially pistachios, have been associated with better glucose control. This nut can modulate the insulin signaling pathway (via PI3K-AKT), favoring glycemic control [46]. However, higher doses (40 g/day or 20% of total energy) may be needed [61].

Indeed, few interventions regarding BN intake have included participants with diabetes, dyslipidemia, or coronary artery disease. Hence, the evidence is limited. In this population, as in others, the effects of daily BN intake seem to be related to Se metabolism. Therefore, people with low plasma Se levels probably benefit more from BN consumption. Further investigations are necessary to understand the pathways involved.

### 3.6. BN Consumption by Subjects Undergoing Hemodialysis

A range of benefits could be observed in patients undergoing hemodialysis who regularly consumed one unit/day of BN (290 µg of Se) for three months. These benefits included enhanced non-enzymatic (increased Se in plasma and erythrocytes) and enzymatic (increased GSH-Px activity in erythrocytes and GPx activity in plasma) antioxidant defense systems, in line with the increased Nrf2 and NQO1 mRNA expression. In addition, the studies also showed improvements in inflammation (reduced Nf-κB mRNA expression, TNF, and IL-6 levels), oxidative stress (reduced malondialdehyde and 8-isoprostanes), and DNA damage (decreased 8-OHdG). There were also improved lipid profiles with an increased HDL-c and decreased atherogenic indices. Furthermore, elevated T3 and free thyroxine (FT4) levels were observed [19,34,35,37] (Figure 3). The same group of researchers in Brazil published all of these results. As Figure 4C presents, subjects undergoing hemodialysis were Se-deficient before BN supplementation (Figure 4).

Dialysis is a treatment for end-stage renal disease caused by chronic kidney disease, a debilitant disease associated with altered immune response, systemic inflammation, and oxidative stress. In addition to high oxidant production, subjects with chronic kidney disease have impaired antioxidative systems, favoring an oxidizing state [62]. These processes are responsible for the metabolic disorders observed in subjects with chronic kidney disease, such as dyslipidemias, who become susceptible to developing atherosclerosis, CVDs, and other complications [63]. Faced with a weakened state of health, subjects undergoing hemodialysis benefited greatly from BN consumption, a food matrix with a valuable nutritional profile, especially antioxidant compounds such as Se.

Although inflammation and oxidative stress are natural mechanisms in the body, they lead to the overproduction of pro-inflammatory mediators and oxidant agents when unregulated. For example, uremic toxins, such as indoxyl sulfate, produced by subjects with chronic kidney disease, are one of the agents responsible for activating NF-κB. Thus, pro-inflammatory cytokines, such as IL-6 and IL-1, are produced, and TNF, oxygen, and reactive hydrogen species feed a vicious cycle of stimulating the production of pro-inflammatory molecules and consequent inflammation exacerbation. Additionally, uremic toxins are linked to malondialdehyde and peroxynitrite production and also advanced glycation end products. These are related to NF-κB activation and a reduction in the activity of the transcription factor Nrf2 and, consequently, the underregulated expression of antioxidant enzymes [62]. These events favor a state of oxidative stress, making DNA susceptible to oxidative damage, as marked by an increase in 8-OH-dG.

Subjects with chronic kidney disease and end-stage renal disease are deficient in Se due to reduced intestinal absorption, loss during hemodialysis, and other causes. This worsens the depleting state of antioxidant agents. Therefore, BN supplementation seems to be interesting, improving the antioxidant, inflammatory, and oxidative stress status. However, 8-isoprostanes, 8-OHdG, IL-6, and TNF levels were significantly increased, whereas Se levels and GPx activity decreased 12 months after the intervention was interrupted [28,30], highlighting the importance of regular BN consumption. In addition, the kidneys participate in thyroid hormones’ metabolism, which tends to be impaired in subjects with chronic kidney disease. As Se participates in thyroid hormone formation, BN supplementation increased T3 and FT4 levels in hemodialysis subjects.

### 3.7. BN Consumption by Subjects with MCIs

Consumption of one unit/day of BN (288.75 µg of Se) for 24 weeks increased Se levels in plasma and erythrocytes, plasma GPx activity, verbal fluency, and construction praxis in older adults with MCIs (Figure 3). In addition, Se status was positively correlated with the Consortium to Establish a Registry for Alzheimer’s Disease (CERAD) score, with higher scores indicating better cognitive performance [16].

Cognitive impairments and neurodegenerative diseases are related to oxidative stress caused by damage to the central nervous system. For this reason, Se deficiency can compromise the redox balance and reduce cognitive performance [64]. The authors have pointed out that most of the sample had Se deficiency at baseline, which seems to be a determining factor in achieving benefits from Se supplementation. However, in a population of older adults with adequate Se intake (85% of all participants), plasma Se levels were not associated with cognitive performance or neurotrophic factors [65].

Genetic factors also influence the responses to BN intake. For example, single nucleotide polymorphisms (SNPs) in genes encoding selenoproteins seem to negatively contribute to the protective effect against oxidative stress. In this sense, in subjects with MCIs, consuming BN daily (288.75 µg of Se) for 24 weeks increased GPx1 mRNA expression in carriers of a variant allele (CT + TT) for rs105045. Conversely, GPx1 mRNA expression was significantly reduced in rs7579 A-carriers and rs3877899 GG carriers [17].

The SNPs also affected SELENOP mRNA expression in MCI patients. While carriers of a variant allele (CT + TT) for rs1050450 showed increased SELENOP mRNA expression after treatment, no differences were observed for CC carriers [17]. Further studies are needed to clarify how polymorphisms would influence Se markers.

Studies on the effect of BN intake on MCIs are scarce. However, the available evidence shows superior benefits in people with low plasma Se values. In addition, Se deficiency can be a risk factor for cognitive impairment and polymorphisms that affect Se metabolism and utilization.

### 3.8. Future Perspectives

Currently, we know that BN consumption benefits human health, and seems to vary according to different routes of action and the health status of the sample evaluated. In addition, these findings refer to BN consumption in a free-living habitual diet without restrictions or adaptations. We emphasized above that individuals on hemodialysis benefited greatly from BN consumption. Many markers were evaluated in this sample of subjects undergoing hemodialysis, which does not mean that other studies did not observe the same benefits. This suggests that more studies are required to evaluate such markers also in other samples, with or without diseases. Considering that caloric restriction is one of the most reputable approaches in the literature for weight management, studies evaluating the effects of nut consumption in the context of caloric restriction are scarce. A recent study showed the superior effects of consuming of a mix of Brazilian nuts (30 g cashew nuts + 15 g BN) with a 500-calorie restricted diet in improving body composition and reducing the soluble adhesion molecule vascular cell adhesion protein 1 (VCAM- 1) [66]. Thus, it would be interesting to conduct studies in this field using BN. In addition, as Se is a powerful antioxidant, studies evaluating longevity markers are of great relevance alongside those examining the intestinal microbiota and their response to adequate Se blood levels compared to inadequate levels. In view of the evidence regarding the interaction between nutrients and the intestinal microbiota, nuts have been investigated as modulators of the microbiota, since it is a food matrix rich in nutrients. However, to date, no study has investigated the effect of BN on the microbiota. Studies comparing the effect of BN consumption with isolated Se supplementation would also be of great interest because of limited evidence in this field.

## 4. Conclusions

Nowadays, scientific evidence supports the health benefits of regular BN consumption in a free-living habitual diet. This seems to vary according to the health status of the subjects, the Se content of the BNs, the Se levels of the participants before the intervention, and the presence of some polymorphisms. The benefits of BN consumption were found in different action pathways.

## Figures and Tables

**Figure 1 foods-11-02925-f001:**
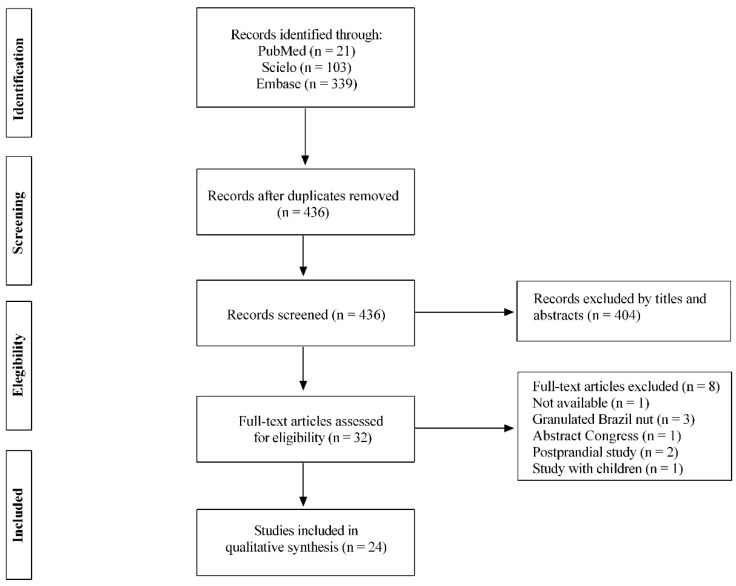
Flowchart of studies included in the systematic review.

**Figure 2 foods-11-02925-f002:**
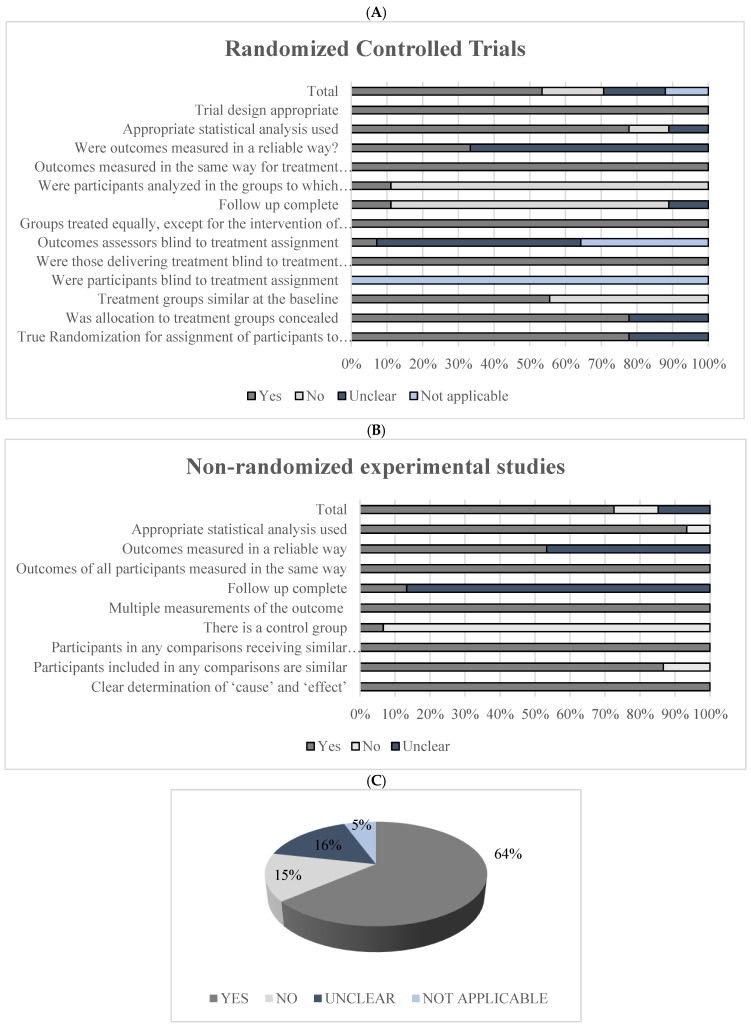
Assessment of studies’ bias risk following the *Joanna Briggs Institute (JBI) Reviewer’s Manual*. (**A**) Risk of bias assessment of randomized clinical trials (*n* = 9); (**B**) Risk of bias assessment of non-randomized clinical trials (*n* = 15); (**C**) Total score.

**Figure 3 foods-11-02925-f003:**
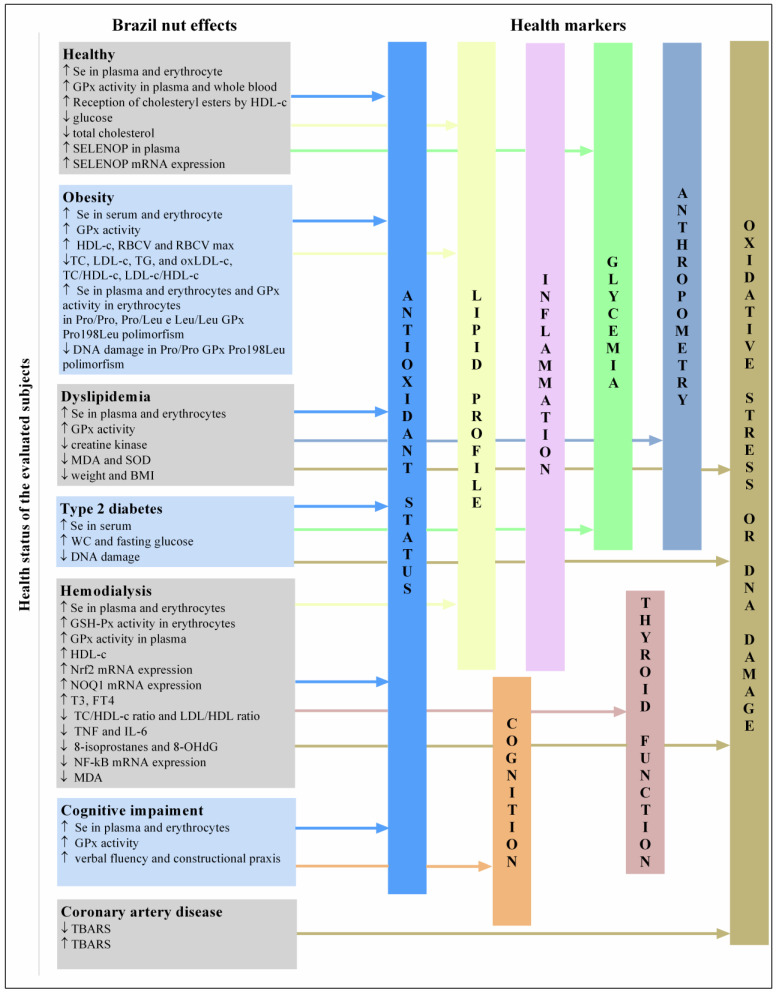
Summary of the effects of Brazil nuts consumption on human health. The effects of regular BN consumption were grouped according to the health status of the subjects evaluated in the studies included in the review. These effects were then linked to a larger category of health markers.

**Figure 4 foods-11-02925-f004:**
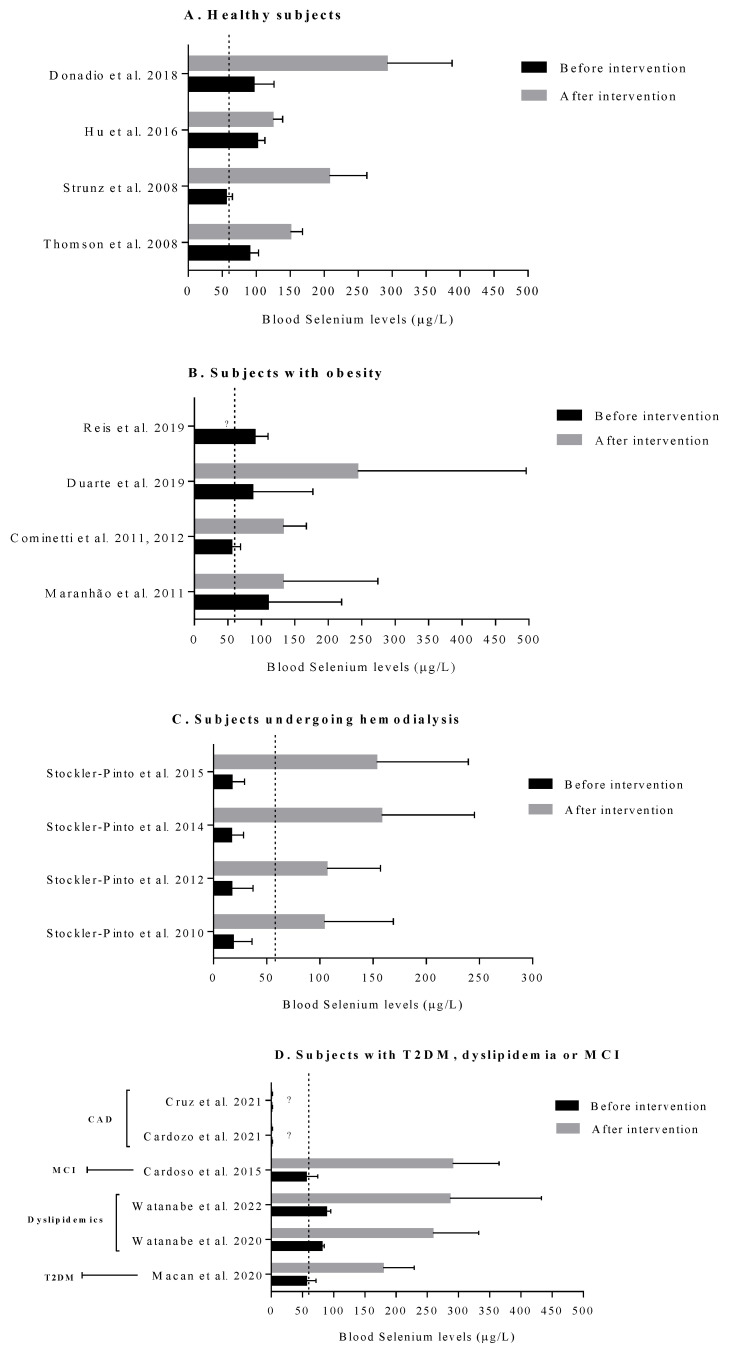
Blood selenium levels (plasma or serum or total blood) before and after Brazil nut supplementation in (**A**). healthy subjects [14,18,24,34], (**B**). subjects with obesity [15,19,20,26,27], (**C**). subjects undergoing hemodialysis [25,28,29,30,31], and (**D**). subjects with type 2 diabetes mellitus (T2DM) or using statins or mild cognitive impairment (MCI) or with coronary artery disease (CAD). ?, not informed [16,17,21,22,35,36,37].

**Table 1 foods-11-02925-t001:** Randomized clinical trials evaluating the effects of Brazil nut consumption on human health (*n* = 9).

Reference Author, Year, Country	Sample Characteristics	Intervention Characteristics	Study Design and Duration	Evaluated Markers	Results
Placebo-controlled clinical trials					
[14] New Zealand	n: 59 healthy subjects G1: Age: 49.5 (SD 8.6) y BMI: 28.2 (SD 5.3) kg/m^2^ G2: Age: 45.6 (SD 11) y BMI: 26 (SD 3.6) kg/m^2^ G3: Age: 42.5 (SD 9.9) y BMI: 25.9 (SD 4.2) kg/m^2^	G1: two units/day of BN (~53 µg of Se) G2: selenomethionine supplement (97.5 ± 11.1 µg of Se) G3: placebo (0.038 ± 0.036 µg of Se)	Randomized, controlled clinical trial 12 weeks	Antioxidant status	G1, G2: ↑ Se and GPx activity in plasma vs. G3 G1: ↑ GPx activity in whole blood vs. G2, G3
[15] Brazil	n: 17 obese female adolescents Age: 15.4 (SD 2) y BMI: 35.6 (SD 3.3) kg/m^2^	G1: 15–25 g/day (three to five units) of BN (108.5 ± 27 µg of Se) G2: placebo (one capsule/day containing lactose)	Randomized, non-blinded pilot trial 16 weeks	Antioxidant status Lipid and glucose metabolism markers Inflammation Anthropometry Oxidative stress	G1: ↓ total cholesterol, triglycerides, and ox-LDL vs. G2 G1: ↑ RBCV vs. G2 G1: ↑ Se and RBCV max vs. baseline ↔ BMI, waist circumference, insulin, glycemia, HOMA-IR, CRP, HDL-c, GPx-3, and 8-epi-PGF2α
Controlled clinical trials					
[16] Brazil	n: 20 older adults with mild cognitive impairment Age: 77.7 (SD 5.3) y	G1: one unit/day of BN (288.75 µg of Se) G2: control	Randomized, controlled clinical trial 24 weeks	Antioxidant status Oxidative stress Cognition	G1: ↑ Se in plasma and erythrocytes vs. G2 G1: ↑ GPx activity in erythrocytes, verbal fluency, construction praxis vs. G2 ↔ ORAC, MDA, CERAD total score, Boston naming test, word list learning test, word list recall
[17] Brazil	n: 20 older adults with mild cognitive impairment Age: 77.7 (SD 5.3) y	One unit/day of BN (288.75 µg of Se)	Secondary analysis of a randomized controlled clinical trial, which evaluated only the group that received BN 24 weeks	Antioxidant status Oxidative stress	↔ Se in plasma and erythrocytes, GPx, ORAC, and MDA activity among genotypes ↑ expression of GPx1 mRNA and selenoprotein P in CT + TT allele carriers for rs1050450 over time ↑ selenoprotein mRNA expression and ↓ GPX1 mRNA expression in A-carriers for rs7579 and GG-carriers for rs3877899
[18] Australia	n: 32 healthy subjects Age: 60 (52–76) y	G1: six units/day of BN (~48 µg of Se) G2: four capsules containing 800 mg of (-) epigallocatechin-3-gallate/day G3: combination of G1 and G2 interventions	Randomized, controlled clinical trial 6 weeks	Antioxidant status Kidney function marker Glucose Inflammation Thyroid function markers Genes/proteins related to the colorectal cancer oncogenesis	G1: ↑ Se and plasma urea vs. G2 G3: ↑ Se and ↓ plasma creatinine vs. G2 ↔ glycemia, CRP, TSH, T3 and T4, mRNA expression of Ac-H3 histones, Ki-67, SELENOP, NF-κB, β-catenin, c-Myc, cyclin D1, and DNNT1 between groups G1 and G2: ↑ SELENOP mRNA expression vs. baseline G1: ↔ DNMT1, NF-κB, c-Myc, and cyclin D1 mRNA expression vs. baseline G1 and G2: ↓ β-catenin vs. baseline
[19] Brazil	n: 55 women with obesity G1: Age: 40.4 (SD 9) y BMI: 34.6 (30.8–37.4) kg/m^2^ G2: Age: 39.4 (SD 9.5) y BMI: 34.8 (33.1–40.2) kg/m^2^	G1: one unit/day of BN (~1261 µg of Se) G2: control	Randomized, controlled clinical trial 8 weeks	Antioxidant status Inflammation Endothelial function markers	G1: ↑ Se in plasma and erythrocytes, GPx1 activity, selenoprotein P, gene expression for selenoproteins, TNF-α, IL-6, IL-10, TLR2, TLR4 and ↓ GPx1 gene expression vs. G2 ↔ CRP, MCP-1, IL-6, IL-10, IL-1 β, TNF-α, IFN-γ, fibrinogen
[20] Brazil	n: 54 women with obesity G1: Age: 40.4 (SD 9) y BMI: 34.9 (SD 4.7) kg/m^2^ G2: Age: 39.4 (SD 9.5) y BMI: 36.6 (SD 6.5) kg/m^2^	G1: one unit/day of BN (~1261 µg of Se) G2: control	Randomized, controlled clinical trial 8 weeks	Antioxidant status	G1: ↑ Se in plasma and erythrocytes, expression of miR-454-3p and miR-584-5p vs. G2
[21] Brazil	n: 42 subjects with coronary artery disease G1: Age: 63.3 (SD 6.7) y BMI: 29.3 (SD 5.6) kg/m^2^ G2: Age: 63.3 (SD 8) y BMI: 28.6 (SD 4.8) kg/m^2^	G1: one unit/day of BN (~290.5 µg of Se) G2: control	Randomized, controlled clinical trial 12 weeks	Lipid metabolism markers Inflammation Oxidative stress	↔ Nrf2, NF-κB, and NQO1 mRNA expression, TC, HDL-c, LDL-c, TG, TC/HDL-c, LDL-c/HDL-c, and TBARS G1: ↓ TBARS (?) vs. G2
[22] Brazil	n: 36 subjects with coronary artery disease G1: Age: 63 (SD 6.7) y BMI: 28.5 (SD 4.5) kg/m^2^ G2: Age: 64.6 (SD 7.2) y BMI: 29.4 (SD 5.3) kg/m^2^	G1: one unit/day of BN (~290.5 µg of Se) G2: control	Randomized, controlled clinical trial 12 weeks	Lipid metabolism markers Inflammation Oxidative stress	↔ PPARβ/δ and NF-κB mRNA expression, CRP, TC, HDL-c, LDL-c, TG, and TNF G2: ↓ TBARS vs. G1

Legend: ↑, increased; ↓, decreased; ↔ unchanged; G, group; SD, standard deviation; BN, Brazil nut; Se, selenium; BMI, body mass index; GPx, glutathione peroxidase; RBCV, red blood cell velocity; oxLDL-c, oxidized low-density lipoprotein cholesterol; HOMA-IR, homeostatic model assessment of insulin resistance; CRP, C-reactive protein; 8-epi-PGF2α, 8-Epi-Prostaglandin F2 Alpha; HDL-c, high-density lipoprotein cholesterol; DNA, deoxyribonucleic acid; ORAC, oxygen radical absorbance capacity; MDA, malondialdehyde; TSH, thyroid-stimulating hormone; T3, triiodothyronine; T4, thyroxine; SELENOP, selenoprotein P; TNF, tumor necrosis factor; interleukin 6; IL-10, interleukin 10; TLR -2, tool-like receptor 2; TLR-4, tool-like receptor 4; MCP-1, monocyte chemoattractant protein 1; IL-1β, interleukin 1β; IFN- γ, interferon gamma; NF-κB, nuclear factor kappa B; Nrf2, nuclear factor erythroid 2-related factor 2; NQO1, NAD(P)H: quinone oxidoreductase 1; TC, total cholesterol; LDL-c, low-density lipoprotein cholesterol; TG, triglycerides; TBARS, thiobarbituric acid reactive substances; PPARβ/δ, peroxisome proliferator-activated receptors β/δ; AST, aspartate transaminase; ALT, alanine transaminase; GGT, gamma-glutamyl transferase; hs-CRP, high-sensitivity C-reactive protein; CERAD, consortium to establish a registry for Alzheimer’s disease.

**Table 2 foods-11-02925-t002:** Non-randomized clinical trials evaluating the effects of Brazil nut consumption on human health (*n* = 15).

Reference	Sample Characteristics	Intervention Characteristics	Study Design and Duration	Evaluated Markers	Results
Controlled clinical trials					
[23] Brazil	n: 25 subjects on hemodialysis G1 (n = 13): Age: 57.1 (SD 12) y BMI: 24.4 (SD 3.2) kg/m^2^ G2 (n = 12): Age: 52 (SD 15.5) y BMI: 26.1 (SD 5.8) kg/m^2^	G1: one unit/day of BN (~5 g with 290.5 µg of Se) G2: control	Controlled clinical trial 12 weeks	Antioxidant status Inflammation Oxidative stress	G1: ↑ mRNA expression of Nrf2, NAD(P)H: quinone oxidoreductase 1 (NQO1) and ↓ mRNA expression of NF-κB vs. G2 G1: ↓ MDA, IL-6 vs. baseline
Uncontrolled clinical trial					
[24] Brazil	n: 15 normolipidemic subjects Age: 27.3 (SD 3.9) y BMI: 23.8 (SD 2.8) kg/m^2^	45 g/day of BN (11 units with 862.65 µg of Se)	Clinical trial 15 days	Antioxidant status Anthropometry Lipid metabolism markers	↑ Se in plasma and reception of cholesteryl esters by HDL-c ↔ weight, total cholesterol, LDL-c, HDL-c, TG, Apo A-I, Apo B, HDL-c diameter, PON 1 activity, % cholesterol, TG, and phospholipid transfer
[25] Brazil	n: 81 hemodialysis patients Age: 52 (SD 15.2) y BMI: 24.9 (SD 4.4) kg/m^2^	One unit/day of BN (~5 g with 290.5 µg Se)	Clinical trial 12 weeks	Antioxidant status	↑ Se in plasma and erythrocytes ↑ GSH-Px activity in erythrocytes (began to be within normal)
[26] Brazil	n: 37 morbidly obese women Age: 34.5 (SD 6.8) y BMI: 45.2 (SD 4.2) kg/m^2^	One unit/day of BN (~290 µg of Se)	Clinical trial 8 weeks	Antioxidant status Glucose Anthropometry DNA damage	↑ Se in plasma and erythrocytes and GPx activity in all genotypes ↓ DNA damage in those with the Pro/Pro genotype. vs. baseline ↑ DNA damage in those with genotype Leu/Leu vs. Pro/Read ↔ weight, BMI, blood glucose
[27] Brazil	n: 37 morbidly obese women Age: 34.5 (SD 6.8) y BMI: 45.2 (SD 4.2) kg/m^2^	One unit/day of BN (~290 µg of Se)	Clinical trial 8 weeks	Antioxidant status Lipid profile markers Glucose Anthropometry	↑ Se in plasma and erythrocytes ↑ GPx activity ↑ HDL-c ↓ TC/HDL-c and LDL-c/HDL-c ratio ↔ weight, BMI, total cholesterol, LDL-c, VLDL-c, TC, fasting glucose
[28] Brazil	n: 21 hemodialysis patients Age: 54.2 (SD 15.2) y BMI: 24.4 (SD 3.8) kg/m^2^	One unit/day of BN (~5 g with 290.5 µg Se)	Clinical trial 12 weeks	Antioxidant status Anthropometry, body fat, Kidney function markers Minerals	↑ Se in plasma ↓ urea nitrogen Follow-up after 12 months: ↓ Se in plasma and urea nitrogen ↔ BMI, body fat, WC, creatinine, minerals
[29] Brazil	n: 40 hemodialysis patients Age: 53.3 (SD 16.1) y	One unit/day of BN (~5 g with 290.5 µg Se)	Clinical trial 12 weeks	Antioxidant status Lipid metabolism markers Inflammation Oxidative stress and DNA damage	↑ Se, GPx activity and HDL-c in plasma ↓ TNF, IL-6, 8-OHdG, 8-isoprostane, LDL-c, Castelli index I and II ↔ total cholesterol, TG
[30] Brazil	n: 29 hemodialysis patients Age: 51 (SD 3.3) y BMI: 23.6 (17.7–40.3) kg/m^2^	One unit/day of BN (~5 g with 290.5 µg Se)	12-month follow-up after 3 months of BN consumption	Antioxidant status Lipid metabolism markers Inflammation Oxidative stress and DNA damage	↓ Se and GPx activity in plasma ↑ TNF, IL-6, 8-OHdG, 8-isoprostane ↔ total cholesterol, TG, LDL-c, HDL-c
[31] Brazil	n: 40 hemodialysis patients Age: 53.3 (SD 16.1) y	One unit/day of BN (~5 g with 290.5 µg Se)	Clinical trial 12 weeks	Antioxidant status Thyroid function markers	↑ Se in plasma, GPx activity, T3 and T4 levels ↔ TSH
[32] Brazil	n: 130 healthy subjects Age: 29.8 (SD 9.2) y BMI: 23.3 (SD 3.3) kg/m^2^	One unit/day of BN (3 to 4 g with ~300 µg of Se)	Clinical trial 8 weeks	Glucose and lipid metabolism markers	↓ glucose at 4 and 8 weeks and total cholesterol at 8 weeks
[33] Brazil	n: 130 healthy subjects Age: 29.8 (SD 9.2) y BMI: 23.3 (SD 3.3) kg/m^2^	One unit/day of BN (3 to 4 g with ~300 µg of Se)	Clinical trial 8 weeks	Antioxidant status	↑ mRNA expression of GPX1 in subjects with genotype in rs713041 ↑ Selenoprotein P mRNA expression in A allele carriers in rs7579 before and after consumption
[34] Brazil	n: 130 healthy subjects Age: 29.8 (SD 9.2) y BMI: 23.3 (SD 3.3) kg/m^2^	One unit/day of BN (3 to 4 g with ~300 µg of Se)	Clinical trial 8 weeks	Antioxidant status	GPx1 activity: ↓ at 4 weeks but did not differ from baseline at 8 weeks GPx3 activity: ↑ at 4 weeks but did not differ from baseline at 8 weeks Se in plasma and erythrocytes: ↑ at 4 and 8 weeks Selenoprotein P: ↑ in 8 weeks
[35] Brazil	n: 60 subjects with type 2 diabetes Men: Age: 62 (SD 9) y BMI: 30.2 (SD 3.2) Women: Age: 66 (SD 8) BMI: 32.6 (SD 4.1)	One unit of BN/day (~3.7 g with 213.67 µg of Se)	Clinical trial 24 weeks	Antioxidant status Anthropometry DNA damage Glucose metabolism markers	↑ Se, waist circumference, glycemia ↓ DNA damage, both basal and cell-induced oxidative damage ↔ BMI, HbA1c
[36] Brazil	n: 32 patients using statins	G1: one unit/day of BN (~5 g with 290 µg of Se) for subjects classified as having high concentrations of creatine kinase G1: one unit/day of BN (~5 g with 290 µg of Se) for subjects classified as having normal creatine kinase concentration	Clinical trial 12 weeks	Antioxidant status Oxidative stress Lipid metabolism marker	G1, G2: ↓ concentrations of protein kinase, MDA, SOD vs. baseline G1, G2: ↑ Se in plasma and erythrocytes, GPx vs. baseline ↔ total cholesterol and mRNA expression of selenoproteins
[37] Brazil	n: 32 patients using statins Age: 50.1 (SEM 7.6) y BMI: 31.1 (SEM 3.8) kg/m^2^	One unit/day of BN (~5 g with 290 µg of Se)	Clinical trial 12 weeks	Antioxidant status Anthropometry Oxidative stress Lipid metabolism marker	↑ erythrocyte GPx activity in all genotypes for the rs1050450 polymorphism in the GPx ↑ erythrocyte GPx activity for those with CC genotype for the rs3877899 polymorphism in the SELENOP and all genotypes for rs7579 polymorphism in the SELENOP ↓ creatine kinase in all genotypes for the rs1050450 polymorphism in the GPx ↓ creatine kinase for those with CC genotype for the rs3877899 polymorphism in the SELENOP and GG genotype for rs7579 polymorphism in the SELENOP

Legend: ↑, increased; ↓, decreased; ↔ unchanged; G, group; SD, standard deviation; BN, Brazil nut; Se, selenium; BMI, body mass index; Nrf2, nuclear factor erythroid 2-related factor 2; NQO1, NAD(P)H: quinone oxidoreductase 1; NF-κB, nuclear factor kappa B; MDA, malondialdehyde; IL-6, interleukin 6; HDL-c, high-density lipoprotein cholesterol; LDL-c, low-density lipoprotein cholesterol; TG, triglycerides; VLDL-c, very-low-density lipoprotein cholesterol; PON 1, paraoxonase 1; Apo-A1, apolipoprotein A-1; Apo B, apolipoprotein B; GSH-Px, glutathione peroxidase; GPx, glutathione peroxidase; TNF, tumor necrosis factor; T3, triiodothyronine; T4, thyroxine; TSH, thyroid-stimulating hormone; DNA, deoxyribonucleic acid; HbA1c, glycated hemoglobin; SOD, superoxide dismutase; 8-OHdG, 8-hydroxydeoxyguanosine; SELENOP, selenoprotein P.

## Data Availability

Not applicable.

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
