# Peer review of "Effects of Regular Brazil Nut (Bertholletia excelsa H.B.K.) Consumption on Health: A Systematic Review of Clinical Trials"

_foods, 2022, doi:10.3390/foods11182925_

Round 1
Reviewer 1 Report
Title: Brazil nuts or Brazil nut? more than one variety?
Abstract: Clear and concise explanation of main findings on Brazil nut usages on health. However, does the scientific name of BN necessary?
Introduction: Too short, more explanation and research gap should be presented on the safety and toxicity of BN
Provide statistical usage of BN in the past 2 years
Material and methods: What was the keywords used during the literature search? Why Web of Science was not used as one of the search engines?
Provides a clear Figure explanation of PRISMA.
Eligibility explain why selected criteria relates with the findings
Figure 3 is low in quality, please revise
Refer to Stockler-Pinto, M. B., Mafra, D., Moraes, C., Lobo, J., Boaventura, G. T., Farage, N. E., ... & Malm, O. (2014). Brazil nut (Bertholletia excelsa, HBK) improves oxidative stress and inflammation biomarkers in hemodialysis patients. Biological trace element research, 158(1), 105-112. in order to improve your discussion
Provide figure for 3.5 and 3.6 to get better readership
Author Response
TITLE: “Effects of regular Brazil nut (Bertholletia excelsa H.B.K.) consumption on health: A systematic review of clinical trials”
Dear reviewer,
We would like to thank you for every comment and suggestion. We appreciate all comments, questions and suggestions to improve the article. We think this is the way to do Science! Each improving the other. It is wonderful.
We made the requested changes and improved the English writing. A revised English proofing certificate has been attached to this file. All changes have been highlighted in red in the text. We hope to meet your expectations.
Thank you in advance.
Yours sincerely,
The authors
Comments and Suggestions for Authors
#1 Title: Brazil nuts or Brazil nut? more than one variety?
Answer: Thank you very much for your inquiry. It's just a "Brazil nut" variety. We corrected the title as suggested.
#2 Abstract: Clear and concise explanation of main findings on Brazil nut usages on health. However, does the scientific name of BN necessary?
Answer: We appreciate your suggestion. We removed the scientific name from the abstract as suggested.
#3 Introduction: Too short, more explanation and research gap should be presented on the safety and toxicity of BN
Answer: Thanks for the suggestion. The introduction of the article was short, but we believe it was objective because more details are discussed throughout the article. We have inserted a text on safety and toxicity as suggested.
#4 Provide statistical usage of BN in the past 2 years
Answer: Thanks for the suggestion. We insert this information into the text as requested.
#5 Material and methods: What was the keywords used during the literature search? Why Web of Science was not used as one of the search engines?
Answer: The keywords used in the searches are listed in Supplementary Table 1 (page 29). We did not use the "Web of Science" database because we considered that the international databases PubMed and EMBASE were sufficient to search for articles within our research question.
#6 Provides a clear Figure explanation of PRISMA.
Answer: Thanks for the suggestion. We inserted a new flowchart based in the correction suggested by Reviewer’s 2. We removed the word “updated” and then changed the flowchart.
#7 Eligibility explain why selected criteria relates with the findings
Answer: Thanks a lot for the suggestion. We have inserted more details into the text as requested.
Figure 3 is low in quality, please revise
Answer: Thanks for the suggestion. We improved the figure. Now it is at 600 dpi.
#8 Refer to Stockler-Pinto, M. B., Mafra, D., Moraes, C., Lobo, J., Boaventura, G. T., Farage, N. E., ... & Malm, O. (2014). Brazil nut (Bertholletia excelsa, HBK) improves oxidative stress and inflammation biomarkers in hemodialysis patients. Biological trace element research, 158(1), 105-112. in order to improve your discussion
Answer: Thank you very much for the suggestion. This article was one of those included in our review and it has already been added to the discussion.
#9 Provide figure for 3.5 and 3.6 to get better readership
Answer: Thanks for the suggestion. Figure 3 illustrates the findings of these two topics and the others included in this review.

Reviewer 2 Report
1. Title “updated”: Compared with previous paper (Ref 2, 4-8), in what areas has it been updated?
2. The review on structure-activity relationship of the compositions in Brazil nuts was insufficient.
3. “Future Perspectives” was insufficient.
4. Introduction: The author should provide information on the planting, processing and use of BN.5. Line 261-266: this paper should review the structure-activity relationship of the compositions (MUFA, polyphenols, and phytochemicals, protein) in Brazil nuts. “For this reason, we believe that most of the benefits achieved from BN intake are related to Se content.” For this sentence, more evidence is needed.
6. As a review, Future perspectives should be provide in detail. In this study, it was too simple. The content is well known to all. The author should propose a good development direction for BN according to the contents of the paper.
Author Response
TITLE: “Effects of regular Brazil nut (Bertholletia excelsa H.B.K.) consumption on health: A systematic review of clinical trials”
Dear reviewer,
We would like to thank you for every comment and suggestion. We appreciate all comments, questions and suggestions to improve the article. We think this is the way to do Science! Each improving the other. It is wonderful.
We made the requested changes and improved the English writing. A revised English proofing certificate has been attached to this file. All changes have been highlighted in red in the text. We hope to meet your expectations.
Thank you in advance.
Yours sincerely,
The authors
Comments and Suggestions for Authors
#1. Title “updated”: Compared with previous paper (Ref 2, 4-8), in what areas has it been updated?
Answer: We appreciate your suggestion. We removed the word “updated” from the title as suggested. This is the first systematic review on the effects of BN consumption on human health. With that in mind, this is not an updated systematic review.
#2. The review on structure-activity relationship of the compositions in Brazil nuts was insufficient.
Answer: We appreciate your affirmation. We have inserted a Table with the composition of Brazil nut provided in each study. In addition, we added the composition provided by the USDA composition table.
#3. “Future Perspectives” was insufficient.
Answer: We appreciate your suggestion. We improved the Future Perspectives as requested.
#4. Introduction: The author should provide information on the planting, processing and use of BN.
Answer: We appreciate your suggestion. We apologize to the reviewer, but this information is already clearly defined in the article by Cardoso et al. 2017. In order not to be repetitive, we decided to reference the article and focus on the subject that the article aims at.
#5. Line 261-266: this paper should review the structure-activity relationship of the compositions (MUFA, polyphenols, and phytochemicals, protein) in Brazil nuts. “For this reason, we believe that most of the benefits achieved from BN intake are related to Se content.” For this sentence, more evidence is needed.
Answer: We appreciate your suggestion. We have inserted a Table with the composition of Brazil nut provided in each study. In addition, we added the composition provided by the USDA composition table.
#6. As a review, Future perspectives should be provide in detail. In this study, it was too simple. The content is well known to all. The author should propose a good development direction for BN according to the contents of the paper.
Answer: We appreciate your suggestion. We inserted more details in this section as suggested.

Round 2
Reviewer 1 Report
All have been answered. Some final suggestions;
1. Add graphical abstract
2. I think having pictures of brazil nut is crucial or other related species would be more appealing to readers.
3. (Introduction) Discuss relation of BN in comparison with other key legumes in food security / underdeveloped legumes. Refer https://doi.org/10.1007/s12571-019-00977-0
Author Response
TITLE: “Effects of regular Brazil nut (Bertholletia excelsa H.B.K.) consumption on health: A systematic review of clinical trials”
Dear reviewer,
We would like to thank you for every comment and suggestion.
We made the requested changes. All changes have been highlighted in red in the text. We hope to meet your expectations.
Thank you in advance.
Yours sincerely,
The authors
Comments and Suggestions for Authors
All have been answered. Some final suggestions;
Question #1: Add graphical abstract
Answer: Thank you for the suggestion. I added the graphical abstract as suggested.
Question #2: I think having pictures of brazil nut is crucial or other related species would be more appealing to readers.
Answer: Thank you for the suggestion. We added the figure as suggested.
Question #3: (Introduction) Discuss relation of BN in comparison with other key legumes in food security / underdeveloped legumes. Refer https://doi.org/10.1007/s12571-019-00977-0
Answer: Thank you for the suggestion. We have inserted the reference in the Introduction section as suggested.

Reviewer 2 Report
OK
Author Response
Dear reviewer,
We would like to thank you for every comment and suggestion. We are pleased to have met your expectations.
Thank you in advance.
Yours sincerely,
The authors